



Geoscientific Model Development (gmd)

# *ConvectiveFoam1.0*: development and benchmarking of a infinite-Pr number solver.

Sara Lenzi[1,2], Matteo Cerminara[3], Mattia de' Michieli Vitturi[3], Tomaso Esposti Ongaro[3], and Antonello Provenzale[2]

[1]Graduate School in Physics and Astrophysics, Torino.
[2]Istituto di Geoscienze e Georisorse, CNR, Pisa.
[3]Istituto Nazionale di Geofisica e Vulcanologia, Sezione di Pisa.

**Correspondence:** Sara Lenzi (sara.lenzi@igg.cnr.it)

**Abstract.** We developed a new fluid-dynamical numerical model, which we call *convectiveFoam*, designed to simulate fluids with very large Prandtl number. First we implemented the high-Pr case, in which advection still acts explicitly, and then the Pr → ∞ version, where the momentum equation becomes diagnostic (that is, without time derivatives) and it is formalized as

an elliptic problem. The new solver, based on a finite volume integration method, is developed on the OpenFOAM platform and it exhibits a good performance in terms of computational costs and accuracy of the results. Scaling properties show a maximum performance around 16000 cells/core, in agreement with other works developed on the same platform. A systematic validation of the solver was performed for both 2D and 3D geometries, showing that *convectiveFoam* is able to reproduce the main results of several iso-viscous cases. This new solver can thus simulate idealized configurations of natural geophysical

convection, such as in the Earth Mantle where $\mathrm{Pr} = 10^{23}$. This solver represents a starting point for general exploration of the behaviour and parameter dependence of several fluid systems of geological interest.

## 1 Introduction

Convection in the Earth mantle is driven by internal heating associated with radioactive decay, cooling at the surface, crustal subduction and interaction between rock and liquid water. Mantle convection in turn affects plate tectonics and the global

carbon cycle. Since our planet is the only known one which shows an active tectonics, several researchers also suggested a link between this phenomenon and Earth habitability. Despite such crucial importance in determining planetary equilibria, however, mantle convection is still only partially understood, owing to the extreme difficulties in examining planetary interior dynamics (Bercovici and Ricard, 2014; O'Neill et al., 2007). To cite a few, the Earth mantle exhibits a viscoelastic nature, behaving like a solid on short time scales but like a fluid on the long ones. This behaviour is synthesized by the Maxwell time:

$$\tau_M = \frac{\eta}{\mu_R} \tag{1}$$

defined as the ratio between dynamic viscosity ($[\eta] = \mathrm{Pa} \cdot \mathrm{s}$) and elasticity (or shear modulus $[\mu_R] = \mathrm{Pa}$). A typical value for the Earth mantle is $\tau_M \approx 10^{10} s$ which relates the evolution to the geological time scales. Also the characteristic spatial scales





(about 3000 km) are a non trivial obstacles to direct geological measurements (Bercovici and Ricard, 2014). For these reasons, indirect methods have usually been employed, such as seismic data analysis or numerical simulations. In particular, for the long

time-scale behaviour ($t \gtrsim \tau_M$), numerical simulations produce diagnostic and prognostic descriptions of geological processes (Khaleque et al., 2015). Furthermore, because of the high viscosity values ($\nu = 10^{18} - 10^{21} \mathrm{m^2 s^{-1}}$), the infinite-Pr number assumption, where $\mathrm{Pr} = \frac{\nu}{\kappa}$, $\nu$ is the kinematic viscosity and $\kappa$ the thermal diffusivity, is usually adopted for the Earth mantle (Bercovici, 2007; Turcotte and Schubert, 2014). Simulating high-Pr regimes poses peculiar numerical difficulties due both to time steps constraints and changes of the mathematical structure of governing equations (Busse, 1979).

This work is a first step towards the exploration of Earth Mantle processes through a model suitable to simulate both high- and infinite-Pr fluid flows. In its first implementation, we will present the model for an isoviscous, single-phase fluid. However, the numerical model is conceived to include future applications with non-Newtonian viscosities (dependent on temperature and pressure), multi-phase and multi-component flows, and for non Cartesian problems, potentially suitable to simulate complex geometries (e.g., slabs or spherical shells) and flows with strong interfaces, needing locally refined numerical meshes.

To develop the new tool for the numerical simulations we adopt the OpenFOAM® (Open Field Operation and Manipulation) toolkit, a free, open source, and widely diffused CFD software. OpenFOAM offers a variety of C++ libraries for Finite Volume (FV) discretisation of partial differential equation systems on a three-dimensional unstructured, colocated mesh. It includes libraries for data manipulation and linear algebra, and a number of numerical solvers for the solution of CFD problems. Thanks to the high level of abstraction, numerical solvers can be developed following pre-built templates. The segregated, iterative

solution method at the base of most pre-built solvers allows the solution of CFD problems with complex, non-linear rheologies, compressible/incompressible regimes, turbulent flows, and the solution of multi-phase, multi-component fluids, withouth increasing the size of the linear systems, thus keeping the computational complexity affordable. OpenFOAM is parallelized with a domain decomposition approach using MPI libraries. It displays satisfactory parallel efficiency up to thousands of cores and offers a variety of pre- and post-processing integrated tools. OpenFOAM is released open-source and is supported by a

broad users and developers community worldwide, making it suitable for the development of community models.

All built-in OpenFOAM solvers are implemented for low-Pr viscous fluids and several rheological laws are already implemented for fluids like water, air, oil, but also for multiphase mixtures of gas, liquids and granular fluids (OpenCFD, 2007). Some extensions of this provided solvers have been recently developed and applied to Earth Science (Cerminara et al., 2016; Dietterich et al., 2017; Rosi et al., 2018). Since an infinite-Pr solver is not provided in the current version of OpenFOAM, we

built a new solver to simulate idealized problems in geological convection. We started exploring high-Pr fluid behaviour, up to that of infinite-Pr number fluids (under the assumption of isoviscosity). For infinite-Pr number fluids, we implemented a brand new solver, which is then validated for 2D and 3D configurations.

The structure of this paper is as follows: a theoretical description of the problem is presented in Sec.2; Sec.3 introduces the numerical setting with an overview on code development. The validation of the new solver is presented in Sec.4. Conclusions

and perspectives are given in Sec.5.





## 2 Model Equations

As in the milestone work on convection done by Rayleigh (Rayleigh, 1916) and Bénard (Bénard and Avsec, 1938) at the beginning of last century, the convection model adopted here is based on classical (incompressible) fluid dynamics equations (Busse, 1979; Chandrasekhar, 1981). The equations for momentum, energy and mass conservation in a rotating frame read:

$$\frac{\partial \boldsymbol{u}}{\partial t} + (\boldsymbol{u} \cdot \nabla)\boldsymbol{u} = -\frac{1}{\rho_0}\nabla p - g\beta\Delta T\hat{\boldsymbol{k}} + \nu\nabla^2\boldsymbol{u} - 2\boldsymbol{\Omega}\times\boldsymbol{u} + \boldsymbol{\Omega}\times(\boldsymbol{\Omega}\times\boldsymbol{r}) \tag{2a}$$

$$\frac{\partial T}{\partial t} + (\boldsymbol{u}\cdot\nabla)T = \kappa\nabla^2 T \tag{2b}$$

$$\nabla\cdot\boldsymbol{u} = 0 \tag{2c}$$

where $\boldsymbol{u} = (u,v,w)$ is the three-dimensional velocity, $\boldsymbol{\Omega} = (0,0,\Omega)$ is the angular velocity, $T$ is the temperature field and other parameters are listed in Table 1. The non-dimensional form is obtained by rewriting Eqs.(2) in terms of relevant units, chosen here as the domain thickness $D$, the diffusive time $\tau_\kappa = \frac{D^2}{\kappa}$, the temperature scale $\Delta T$ and the velocity scale $\frac{\kappa}{D}$. The equations thus become:

$$\frac{1}{\mathrm{Pr}}\left[\frac{\partial \boldsymbol{u}'}{\partial t'} + (\boldsymbol{u}'\cdot\nabla)\boldsymbol{u}'\right] = -\nabla p' + \mathrm{Ra}T'\hat{\boldsymbol{k}} + \nabla^2\boldsymbol{u}' - \frac{1}{\mathrm{Ek}}\hat{\boldsymbol{k}}\times\boldsymbol{u}' \tag{3a}$$

$$\frac{\partial T'}{\partial t'} + (\boldsymbol{u}'\cdot\nabla)T' = \nabla^2 T' \tag{3b}$$

$$\nabla\cdot\boldsymbol{u}' = 0 \tag{3c}$$

where primed variables are non-dimensional. In Eqs.(3), $\mathrm{Pr} = \frac{\nu}{\kappa}$ and $\mathrm{Ra} = \frac{g\beta\Delta T D^3}{\nu\kappa}$ are the Prandtl and Rayleigh numbers, representing respectively the ratio between convective and conductive heat transfer and the ratio between kinematic and thermal heat transport. The Ekman number, $\mathrm{Ek} = \frac{\nu}{2\Omega D^2}$, indicates the ratio between Coriolis and viscous effects. Even if Earth mantle

| Sym | Parameter | Value | Units |
|---|---|---|---|
| $g$ | Gravitational acceleration | 9.81 | $\mathrm{m\,s^{-2}}$ |
| $\rho_0$ | Mean density | 4000 | $\mathrm{kg\,m^{-3}}$ |
| $\beta$ | Thermal expansion coefficient | $2\cdot 10^{-5}$ | $\mathrm{K^{-1}}$ |
| $\kappa$ | Thermal diffusivity | $7.5\cdot 10^{-7}$ | $\mathrm{m^2\,s^{-1}}$ |
| $\nu$ | Kinematic viscosity | $2.5\cdot 10^{17}$ | $\mathrm{m^2\,s^{-1}}$ |
| $\Delta T$ | Temperature difference | 1500 | K |
| $\Omega$ | Angular velocity | $7\cdot 10^{-5}$ | $\mathrm{rad\,s^{-2}}$ |
| $D$ | Characteristic length scale | $3\cdot 10^6$ | m |

**Table 1.** Physical parameters typically used in simulations of Earth Mantle convection (Ricard, 2007).

parameters are strongly dependent on pressure and temperature, here we assume constant parameters, fixed as in Ricard (2007). This leads to constant values for Pr and Ra. Moreover, since $\mathrm{Ek} \approx 10^{12}$, the $\mathrm{Pr}\to\infty$ assumption holds. In this limit, Eqs.(3)





become (removing primes for simplicity):

$$\nabla p - \mathrm{Ra}T\hat{\boldsymbol{k}} - \nabla^2 \boldsymbol{u} = 0 \tag{4a}$$

$$\frac{\partial T}{\partial t} + (\boldsymbol{u} \cdot \nabla)T = \nabla^2 T \tag{4b}$$

$$\boldsymbol{\nabla} \cdot \boldsymbol{u} = 0. \tag{4c}$$

where an elliptical PDE governs the dynamics of the velocity field. In this new problem, the velocity field is strictly dependent
on the temperature and pressure fields, since acceleration has been neglected together with the nonlinear inertial term. Never-
theless, the advective term in the temperature equation still acts as a source of nonlinearity. It follows that analytical solutions
are not available and a numerical approach is warranted.

## 3  Numerical Methods

The problem described by Eqs.(4) is solved with OpenFOAM. OpenFOAM uses the FV method discretising the integral form
of governing equations over each control volume (CV). Each CV is indentified by a polyhedral cell and all field values are
calculated in the centre of such cell (i.e. colocated) (Jasak, 1996). Once the mesh is defined, the surface fluxes are reconstructed
using the Gauss theorem. This requires an interpolation step to infer the surface value from the volume one. To this aim
OpenFOAM provides a set of pre-defined interpolation rules (Ferziger and Peric, 2012). In the following we briefly introduce
two rules adopted in our work.

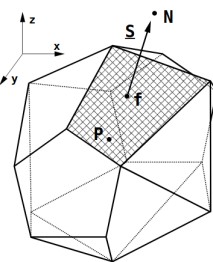

**Figure 1.** Sketch of a control volume from Jasak (1996). Each field is calculated in the centre of a polyhedral cell ( **P**). **N** is the centre of the
nearby cell and **f** the point on the surface on which the field is reconstructed by interpolating between **P** and **N**.

The first is the *Linear* method, also known as *Central Differencing* (*CD*), which reconstructs the field on the surface **S**,
assuming its linear variation between the two points **P** and **N**, located in the centre of two nearby cells. The field value in point
**f** is calculated as:

$$\varphi_f = f_x \varphi_P + (1 - f_x)\varphi_N \tag{5}$$

where $f_x$ is defined as the ratio between $\overline{fN}$ and $\overline{PN}$. *CD* is second order accurate on all kind of meshes (Ferziger and Peric,
2012) even if affected by some non physical oscillations in the solution of convection-dominated problems (Jasak, 1996).





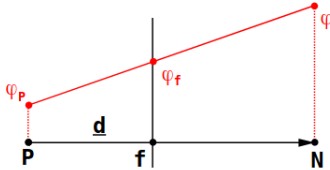

**Figure 2.** Sketch of the 1D version of the linear method from Jasak (1996). The value of the $\varphi$ field on the face passing through **f** is reconstructed by interpolating the values on neighbours cells ($\varphi_P, \varphi_N$) and weighted by $f_x$.

The second rule, the *LimitedLinear* interpolation, removes the oscillations which are tipically associated with second-order schemes (Sweby, 1984). The *Total Variation Diminishing* (*TVD*) criterion was introduced to characterise such oscillation-free flux-limited schemes (Harten, 1983).

OpenFOAM uses a general approach, the *segregated method*, to solve systems of coupled equations as Eqs.(4) (Patankar and Spalding, 1972; Issa, 1986). At each integration step, equations are solved separately for each variable with an iterative procedure, in order to enforce the coupling, until a prescribed global residual level is achieved. Non-linear differential equations are linearized before discretisation and the non-linear terms are lagged (Jasak, 1996). The segregated solution approach consists of two main phases: the PISO (Pressure Implicit with Splitting of Operator) algorithm (Issa, 1986) which solve the momentum equations enforcing the incompressibility constrain, and the SIMPLE (Semi-Implicit Method for Pressure Linked Equations) iterative solver, used to improve the numerical solution of non-linear equations and to enforce the coupling of the momentum and energy equations.

To solve an unsteady, incompressible Navier Stokes problem as in Eqs.(3), OpenFOAM solvers merge the PISO and the SIMPLE procedures into the PIMPLE algorithm. Accordingly, an elliptic equation for the pressure field is derived by a combination of the continuity and momentum equations (Patankar, 1981). The integration procedure for each time step is described by the following (Jasak, 1996):

1. Calculate the time step value based on CFL condition;

2. Momentum predictor: discretize and solve the momentum equation (2a) for the velocity, with pressure from previous iteration;

3. Discretize and solve the temperature equation using the guessed velocity field from step 2;

4. Discretize and solve the pressure equation with the predicted velocity field;

5. Update the velocity field with the new pressure field;

6. PISO loop: repeat from step 4 for a prescribed number of iterations;

7. PIMPLE loop: repeat from step 2 for a prescribed number of iterations or until residual constraints are satisfied;





As part of this work we extended the built in *buoyantBoussinesqPimpleFoam* solver, designed to solve heat transfer prob-
lems, to approach the infinite-Pr number problem. First, the overall structure of the solver has been modified by moving
the temperature equation, originally solved outside the PISO loop, within it. A flowchart comparing the structure of the two
solvers is illustrated in Fig.3 . We compared the performances of the two solvers by monitoring the number of PIMPLE iter-
ations required to converge, that is when residuals, for a given variable, fall under a given threshold. Residuals are defined by
substituting the current solution into the equation and taking the (normalized) magnitude of the difference between the left and
right hand sides of equation members (OpenCFD, 2007).

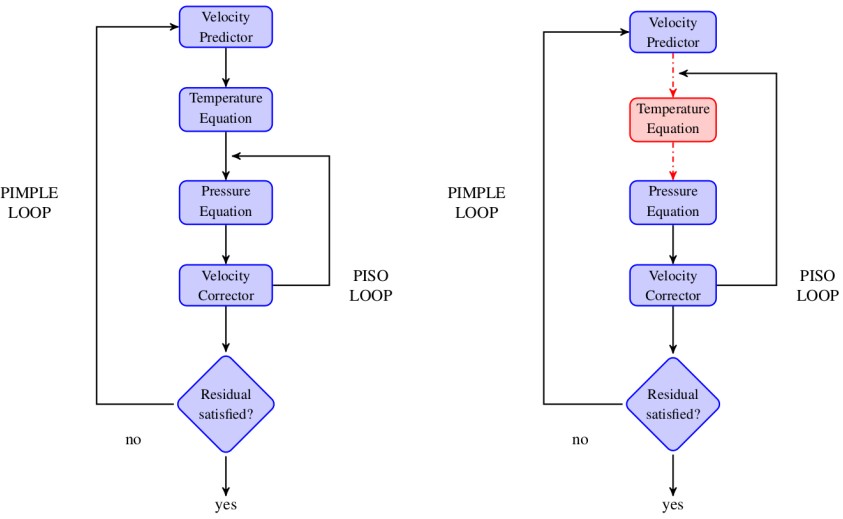

**Figure 3.** Flowcharts of the integration procedure: velocity, temperature and pressure fields are sequentially discretized and solved, then
coupled though the PISO and PIMPLE loops. Left: the old solver *buoyantBoussinesqPimpleFoam*; right: the new solver *convectiveFoam*.
Computation of temperature, originally done outside the PISO loop, has been moved into the loop.

|  | Simulated Time | Execution Time | TOT PIMPLE | Aver.PIMPLE |
|---|---|---|---|---|
| **Teq. IN** | $10^{-2}\tau_\kappa$ | 75 s | $3.1 \cdot 10^3$ | $2.28 \pm 0.26$ |
| **Teq. OUT** | $10^{-2}\tau_\kappa$ | 239 s | $1.3 \cdot 10^4$ | $9.56 \pm 19.53$ |

**Table 2.** Execution time, number and average number of PIMPLE iterations for the two solvers for a simulated time of $10^{-2}\tau_\kappa$ (transient
regime). the $Teq.IN$ solver is about 3 time faster than $Teq.OUT$.


Results show that moving the temperature equation within the PISO loop drastically decreases the computation time. This
fact is quantified in Tab.2 where we compare some representative data, as the total and the average number of final PIMPLE
iterations and the execution time for each solvers, in the case of a $64^2$ grid points domain, for the initial part of simulation
(until $10^{-2}$ thermal time). Analogous results are obtained for longer simulations, until one thermal time: also in this case the
total and the average number of PIMPLE iterations, as the execution time, are compared and reported in Tab.3 (see Fig.4 for





| | Simulated Time | Execution Time | TOT PIMPLE | Aver.PIMPLE |
|---|---|---|---|---|
| **Teq. IN** | $\tau_\kappa$ | $8.0 \cdot 10^3$ s | $3.5 \cdot 10^5$ | $2.75 \pm 0.19$ |
| **Teq. OUT** | $\tau_\kappa$ | $3.8 \cdot 10^4$ s | $2.3 \cdot 10^6$ | $17.99 \pm 56.43$ |

**Table 3.** Execution time, number and average number of PIMPLE iterations for the two solvers for a simulated time of $\tau_\kappa$. The performance is comparable with that of the transient regime: the $Teq.IN$ solver is about 5 time faster than $Teq.OUT$.

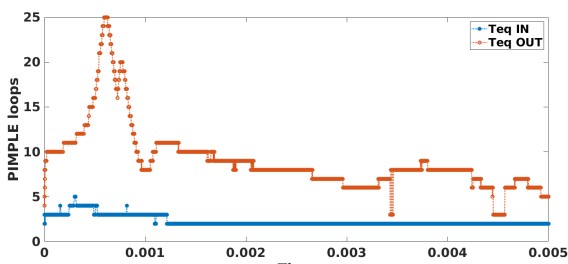
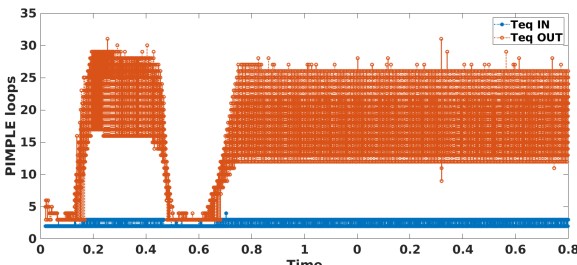

**Figure 4.** Number of PIMPLE iterations as function of time. Left: transient regime ($10^{-2}\tau_\kappa$), right: statistically stationary regime ($\tau_\kappa$). Required loops for $T_{eq}IN$ (blue) are lower than those for $T_{eq}OUT$ (orange) in both cases.

a graphical representation). We concluded that the $T_{eq}IN$ solver performance is clearly improved respect to that of $T_{eq}OUT$ for both the transient and the statistically stationary regime.

A more detailed representation of pressure and temperature residuals behaviour is shown in Fig.5 for a fixed time step. Each PIMPLE loop groups a fixed number of four points which are the initial residuals at every PISO loop. This is valid for the
pressure field, which is always calculated into the PISO loop, but only for $T_{eq}IN$ temperature field. Looking at the pressure panel, the first two PISO loops of each PIMPLE loop result the more important for convergence (cf. 2nd and 3rd point of each group). The same is valid for PIMPLE loops: only two cycles are required by $T_{eq}IN$ solver to converge against the six required by $T_{eq}OUT$. This behaviour, in agreement with results of Tabs.(2,3), confirms that an average value of 2-3 PIMPLE loops is sufficient, for the $T_{eq}IN$ solver, to reach the prescribed convergence.

## 3.1 Implementation of the Pr $\to \infty$ case.

The central goal of this work is to reproduce the infinite-Pr structure. As introduced, Eqs.(3) reduce to the elliptical problem:

$$\nabla p + \mathrm{Ra}T\hat{\boldsymbol{k}} - \nabla^2 \boldsymbol{u} = 0 \qquad . \tag{6}$$

Hyperbolic and elliptic equations require different numerical strategies to be solved. However, to improve the convergence of the linear system solution, we want to maintain the original hyperbolic structure of the solver. This requires the presence of a
time derivative, to preserve the diagonally dominant structure of the matrix resulting from the numerical discretization of the momentum equation (Moukalled et al., 2016). With this aim, we introduced an explicit artificial time derivative on the left hand side of Eq.(6), designed so that it does not affect the result when convergence is reached. This derivative is computed within

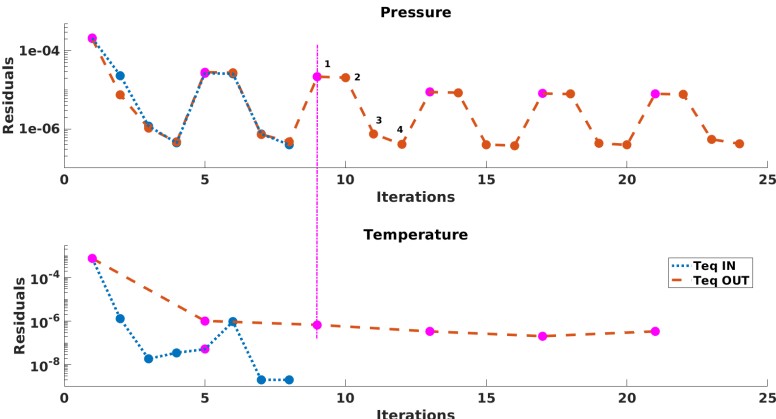

**Figure 5.** Initial residuals for a fixed time step ($t = 0.0098\tau_\kappa$) for the two solvers $T_{eq}IN$ (blue) and $T_{eq}OUT$ (orange). The initial residuals are calculated before the linear system is solved.

TOP panel: pressure residual show a periodicity every 4 points. Each set of consecutive points represents a PIMPLE loop, while each point of the same set (e.g., those market from 1 to 4) is a PISO iteration. The first point of each set (magenta), highlights the initial residual of each PIMPLE loop.

BOTTOM panel: temperature field residuals are evaluated more frequently in the $T_{eq}IN$ configuration, since the temperature equation has been moved into the PISO loop, while, in $T_{eq}OUT$, residuals are computed only at every PIMPLE loop. As shown, $T_{eq}IN$ requires a lower number of PIMPLE loops. In particular, two PIMPLE loops are sufficient to converge.

Is also clear that, in both cases, the first two PISO loops are the main steps to reach convergence.

the PISO loop as:

$$\frac{\Delta \boldsymbol{u}_n}{\Delta t} = \frac{\boldsymbol{u}_n^m - \boldsymbol{u}_n^{m-1}}{\Delta t} \qquad \text{with} \qquad n, m \in \mathbb{N} \tag{7}$$

where $n$ indicates the n-th PIMPLE iteration and $m$ the m-th PISO iteration. Once convergence is reached, $|\boldsymbol{u}_n^m - \boldsymbol{u}_n^{m-1}| < \epsilon$, with $\epsilon$ the prescribed threshold, the auxiliary time derivative of Eq.(7) goes to zero recovering a steady problem.

It is worth noticing that in the Boussinesq approximation the density is held constant in the unsteady and in the advective terms but not in the gravity one, where it is assumed to depend on temperature (Oberbeck, 1879). The Boussinesq OpenFOAM solver defines two auxiliary variables in order to avoid errors in calculating the buoyant term on non-orthogonal and distorted

meshes (OpenCFD, 2007):

$$\rho_\kappa = 1 - \beta(\tilde{T} - T_{ref}), \qquad\qquad p_{rgh} = p - g\rho_\kappa z \tag{8}$$

and it solves the momentum equation with respect to the effective pressure $p_{rgh}$. Since in this work a Cartesian mesh is used, we explicitly rewrote the buoyancy term defining $\rho$ as the sum of a mean value and a perturbation:

$$\boldsymbol{g}\frac{\rho}{\rho_0} = \boldsymbol{g}\left[1 + \frac{\tilde{\rho}}{\rho_0}\right] = \boldsymbol{g}\left[1 - \beta(\tilde{T} - T_0)\right] \tag{9}$$





where a linear relation between temperature and density is assumed with $T_0, \rho_0$ are reference values.

Similarly to the spatial scheme, OpenFOAM provides several temporal discretization strategies. Among them we selected the following (using OpenFOAM definitions (Ferziger and Peric, 2012)):

– *Euler scheme* (first order, implicit), defined as :

$$\frac{\partial f(n)}{\partial t} = \frac{f(n) - f(n-1)}{\Delta t} \tag{10}$$

– *Backward scheme* (second order, implicit):

$$\frac{\partial f(n)}{\partial t} = \frac{\frac{3}{2}f(n) - 2f(n-1) + \frac{1}{2}f(n-2)}{\Delta t}. \tag{11}$$

indicating with $n$ the time step and with $\Delta t$ the time step width.

In the following section, all the previous features are tested, in order to validate the new solver.

## 4   Solver Benchmarking

Our solver has been tested considering both 2D and 3D cases. For the first, we simulated numerically the 2D results reported by Whitehead et al. (2013) for increasing Pr. For the second, instead, we referred to Sotin (1999) in which the infinite-Pr configuration is studied with varying Ra number.

### 4.1   2D Benchmark

We replicated the same conditions of Whitehead et al. (2013): the $xz$-squared domain has fixed BCs for bottom and top temper-
ature ($T = 0$ at $z = 1, T = 1$ at $z = 0 \wedge \frac{1}{4} \leq x \leq \frac{3}{4}$) and zero normal gradient boundary condition at $x = 0$ and $x = 1$. Velocities BCs are free slip on every side of the domain. A $128 \times 128$ grid has been used with the exception of the resolution study where different choices are shown. In Whitehead et al. (2013), the dynamics of a convective process is simulated in the case of a fixed Ra $= 10^6$ number and varying Pr $\in [1, \infty)$. Fig.6 shows the evolution of temperature field for Pr $= 1, 10, 10^2, 10^3, \infty$. The process evolves through four different stages: plume formation, the plume head reaching the top of the domain, the head spreading
and turning back to the bottom. These phases correspond, for Pr $= 10^3$, at times $t = 12.4 \cdot 10^{-4}$, $t = 27 \cdot 10^{-4}$, $t = 50 \cdot 10^{-4}$ and $t = 1000 \cdot 10^{-4}$ (Fig.6, panel d) and all times are in unit of the non-dimensional diffusive time $\tau_\kappa$. The dynamics varies at different values of Pr and the new solver is able to reproduce all of them (Fig.6, panels a-e) even for longer times when the plume starts to oscillate (Fig.6, panel f).

All simulations have been run on the Laki cluster at INGV, Sezione di Pisa ($2 \times 8$-cores Intel Xeon 2.40GHz).

### 4.1.1   Methodology

As in Whitehead et al. (2013), we fixed Ra $= 10^6$, but we focused only on Pr $= 10^3$ and Pr $\rightarrow \infty$ cases, setting the residual threshold to $3 \cdot 10^{-7}$. We performed our analysis comparing temperature and velocity profiles and streamfunction time evolution





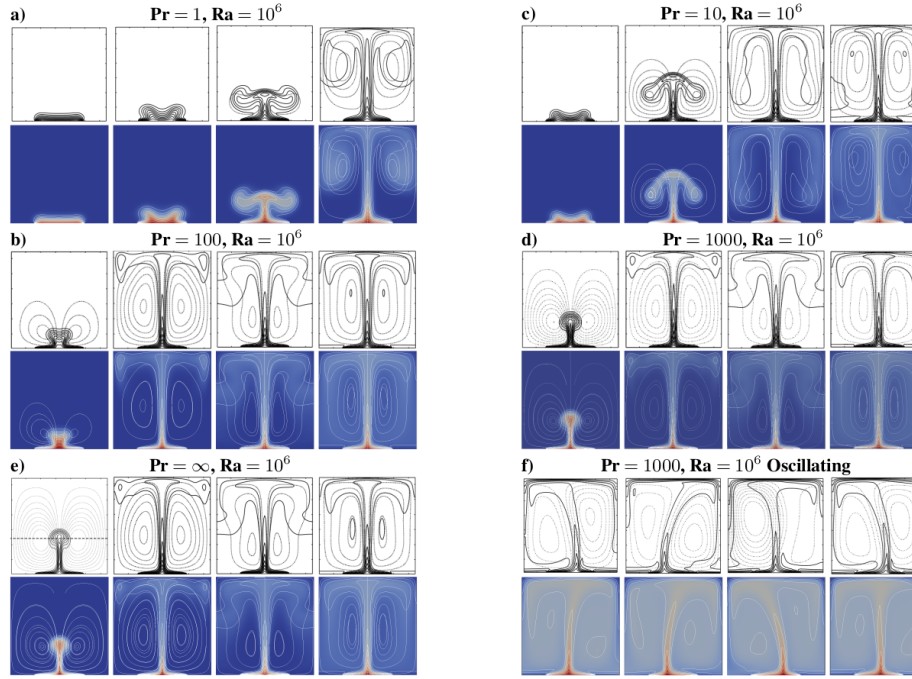

**Figure 6.** Panels a)-e): Snapshots of the temperature field at different time steps for the original work in *Whitehead (2013)* (black and white) and for the new solver *convectiveFoam* (colours). Each panel shows, from left to right, snapshots at: $t = 12.4 \cdot 10^{-4}$, $t = 27 \cdot 10^{-4}$, $t = 50 \cdot 10^{-4}$, $t = 1000 \cdot 10^{-4}$ in unit of the diffusive time $\tau_\kappa$. Panel f): Snapshots of the oscillating temperature field at different time steps (bottom, right). From left, snapshots are taken at: $t = 1810 \cdot 10^{-4}$, $t = 1820 \cdot 10^{-4}$, $t = 1830 \cdot 10^{-4}$, $t = 1840 \cdot 10^{-4}$, in unit of the non-dimensional time $\tau_\kappa$. For each simulation Ra $= 10^6$.

against the originals. The study has been done for three different solvers: the original solver *buoyantBoussinesqPimpleFoam*, the new solver for finite-Pr numbers *convectiveFoam* and for infinite-Pr numbers *convectiveFoamInf*. For each solver, a sys-
tematic analysis has been performed on both short and long term behaviour of the process. In both cases several choices of resolution, algorithm or numerical schemes have been tested, in order to test the sensitivity of the results.

**Resolution**

We ran simulations increasing the number of grid points selecting the following configuration: $64 \times 64$, $128 \times 128$, $256 \times 256$. The main scope of this study is to explore how the resolution affects results, since higher resolution allow to resolve more
detailed fluid structures. We led this analysis following the same approach adopted in Whitehead et al. (2013) where increasing resolution provides different results.





**Numerical Scheme**

FV method requires interpolation rules to reconstruct face fields values from the volume ones. We tested the *Linear* and *LimitedLinear* spatial schemes implemented in OpenFOAM (OpenCFD, 2007). Concerning temporal domain, instead, we tested
OpenFOAM's Euler and Backward schemes. We performed several attempts combining spatial and temporal interpolation rules as follows:

  – *Linear + Euler* (LE);

  – *LimitedLinear + Euler* (LLE);

  – *Linear + Backward* (LB);

– *LimitedLinear + Backward* (LLB);

We used the LB/LLB schemes in conjunction with *convectiveFoam* solver (CLB/CLLB for brevity) and with the original *buoyantBoussinesqPimpleFoam* solver (BLB/BLLB). The same naming convention has consistently extended to the LE and LLE schemes.

**Algorithm**

Algorithm analysis was performed by varying the number of inner PISO loops (which enforces incompressibility) and outer PIMPLE loops (which enforces the coupling with the temperature equation). We named each configuration adopting the *pXpY* convention, where pX stands for X PISO loops and pY for Y PIMPLE loops. Since we observed that a p2p2 configuration is sufficient to reach the satisfactory residual, we focused on a limited number of PISO/PIMPLE iterations. In the following, only a selection of significant results is discussed, since some PISO/PIMPLE combinations, (p1p3 and p3p1), were computationally
unstable.

**4.1.2    Analysis of the Transient Regime**

In this section we report, at different times temperature (Fig. 7) and velocity (Fig. 8) profiles related to the transient scenario comparing them with the originals. To compare the time-dependent behaviour, we also compare the value of the streamfuntion as a function of time (Fig. 9). Figs.7-9 show results from the resolution study performed with the momentum predictor, the
LB scheme and the p2p2 algorithm. Results are in agreement with Whitehead et al. (2013), confirming that the $128 \times 128$ resolution is enough to reproduce all cases, while a $64 \times 64$ grid is insufficient, especially in reproducing the first time steps.

   The results for the different tests on numerical schemes in Figs.10-12 indicate an overall scheme independence, with some discrepancies around $t = 15 \cdot 10^{-4}$, when the velocity and the temperature fields exhibit a highly dynamic behaviour. In this time steps, LB and LLB provide the closest match. The same is valid for the horizontal profile of vertical velocity at $t = 1000 \cdot 10^{-4}$
(Fig.11). For all simulations we used the momentum predictor, a $128 \times 128$ resolution and the p2p2 algorithm.



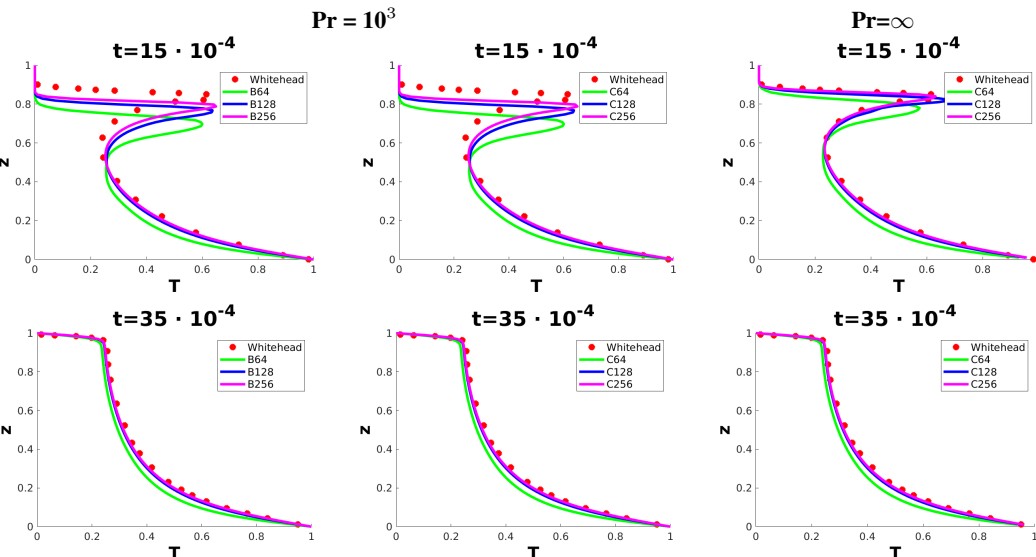

**Figure 7.** Vertical temperature profiles for two different times (upper and lower panels) and for different resolutions: $64 \times 64$ (green), $128 \times 128$ (blue), $256 \times 256$ (magenta). Profiles are calculated at $x = 0.5$. The two left columns report the *buoyantBoussinesqPimpleFoam* (original) and the *convectiveFoam* solver (new) for $Pr = 10^3$; the right column reports the *convectiveFoamInf* results for $Pr \rightarrow \infty$. Benefits of increasing resolution are more appreciable for highly dynamically variable times ($t = 15 \cdot 10^{-4}$ when the plume reaches the top) and less evident at the latest times ($t = 35 \cdot 10^{-4}$).

Concerning algorithm, for all simulations, we used the momentum predictor, the LB scheme and a $128 \times 128$ resolution (Figs.13-15). The analysis of the results suggests p2p2 algorithm as a good compromise both in terms of computational time and accuracy. From the analysis reported above varying resolutions, algorithms and schemes, we selected a reference configuration aimed to develop further studies. In particular, the final configuration has: *Linear* spatial interpolation scheme, *Backward* temporal scheme, $128 \times 128$ spatial resolution and p2p2 algorithm. We remark this is not the more accurate setting but the best compromise between simulation time and accuracy.



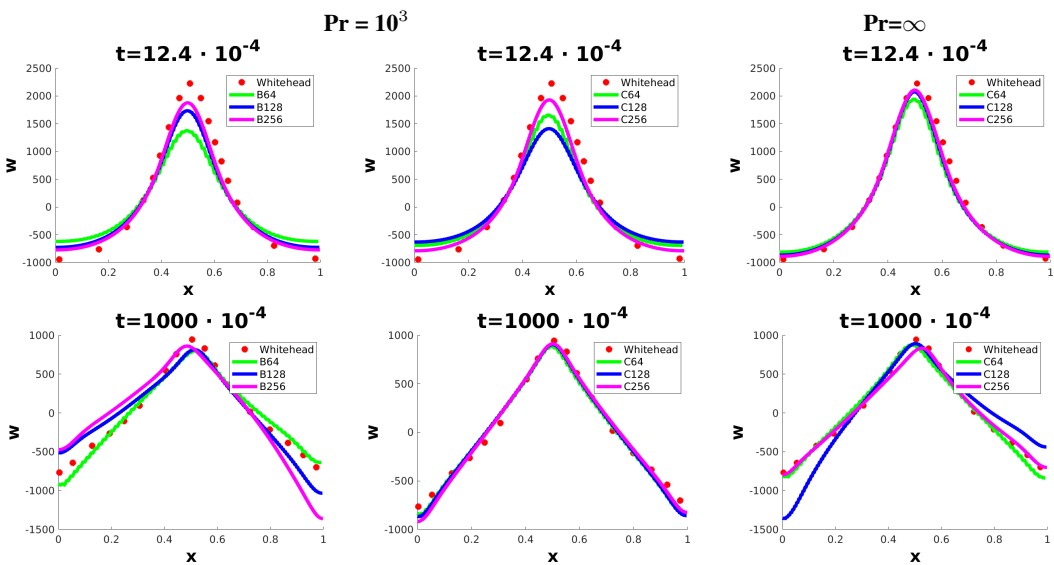

**Figure 8.** Horizontal profiles of vertical velocities at different resolution. Left columns report *buoyantBoussinesqPimpleFoam* and *convectiveFoam* for $Pr = 10^3$; the right column shows the results of *convectiveFoamInf* ($Pr \to \infty$). Profiles are taken at $z = 0.5$ for $t = 1000 \cdot 10^{-4}$, in the middle of the head of the plume for $t = 12.4 \cdot 10^{-4}$. Changing the resolution affects the results for $t = 12.4 \cdot 10^{-4}$, $t = 1000 \cdot 10^{-4}$ for the *buoyantBoussinesqPimpleFoam* and *convectiveFoam* finite-Pr solvers.

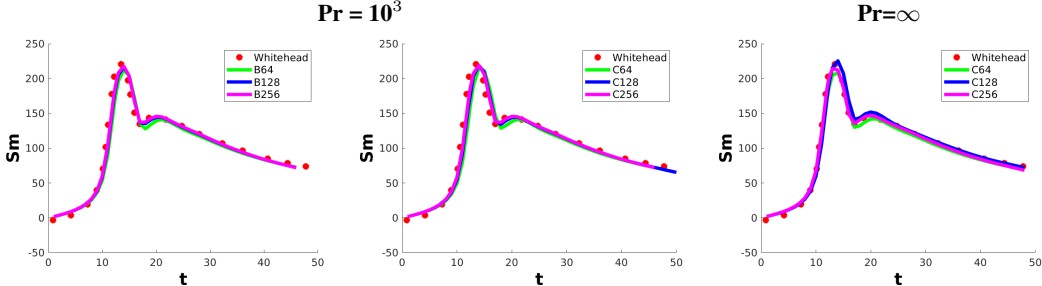

**Figure 9.** Streamfunction evolution for different resolutions, $64 \times 64$ (green), $128 \times 128$ (blue), $256 \times 256$ (magenta), in the interval $[0 - 50] \cdot 10^{-4}$ (in unit of the non-dimensional time $\tau_\kappa$). From the left: *buoyantBoussinesqPimpleFoam*, *convectiveFoam*, *convectiveFoamInf*.



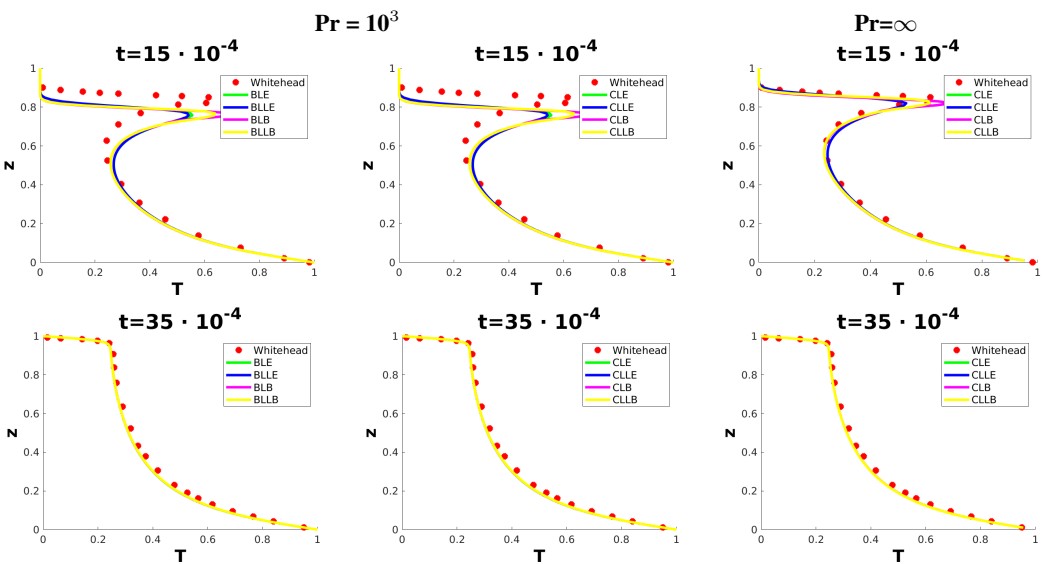

**Figure 10.** Vertical temperature profiles for different schemes at two different times (upper and lower panels): LE (green), LLE (blue), LB (magenta) and LLB (yellow). Profiles are calculated at $x = 0.5$. Left two columns: *buoyantBoussinesqPimpleFoam* and *convectiveFoam* for Pr $= 10^3$. Right column: *convectiveFoamInf* for Pr $\to \infty$. With the only exception of $t = 15 \cdot 10^{-4}$, all times show a good agreement between the different schemes. Note that for the infinite-Pr case, the LB scheme reproduces well also the highly dynamic period $t = 15 \cdot 10^{-4}$.

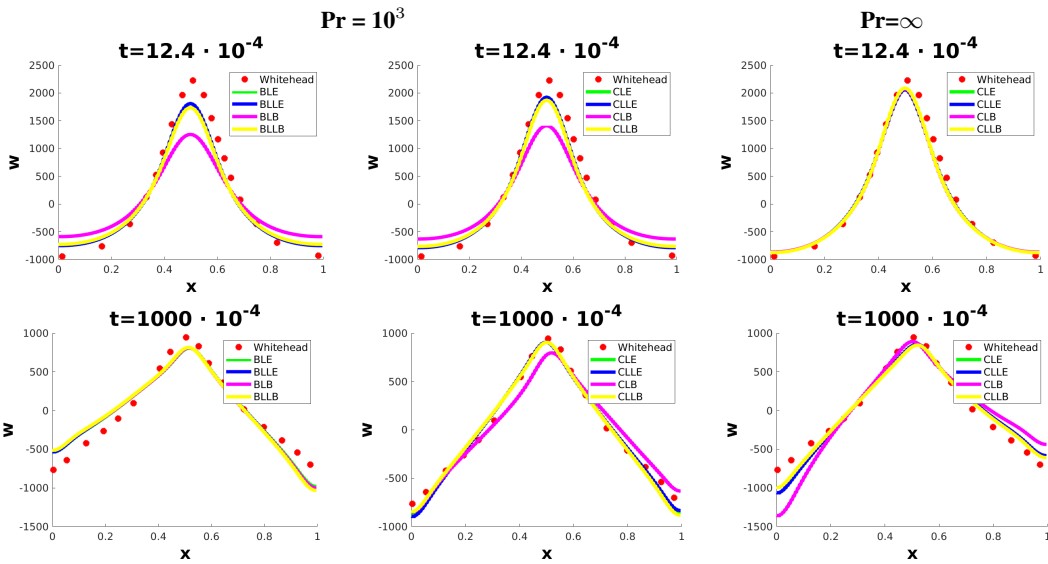

**Figure 11.** Horizontal profiles of vertical velocity for the different schemes for two different times. Profiles are taken at $z = 0.5$ for $t = 1000 \cdot 10^{-4}$, in the middle of the head of the plume for $t = 12.4 \cdot 10^{-4}$. Other details as in Fig.8. In this case, the LB, LLE and LLB schemes provide similar results.



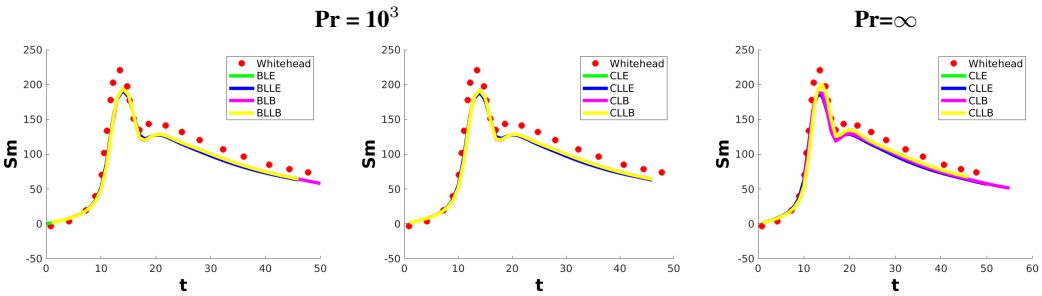

**Figure 12.** Streamfunction evolution for the different schemes: LE (green), LLE (blue), LB (magenta) and LLB (yellow) in the interval $[0 - 50] \cdot 10^{-4}$. From left: *buoyantBoussinesqPimpleFoam*, *convectiveFoam*, *convectiveFoamInf*.

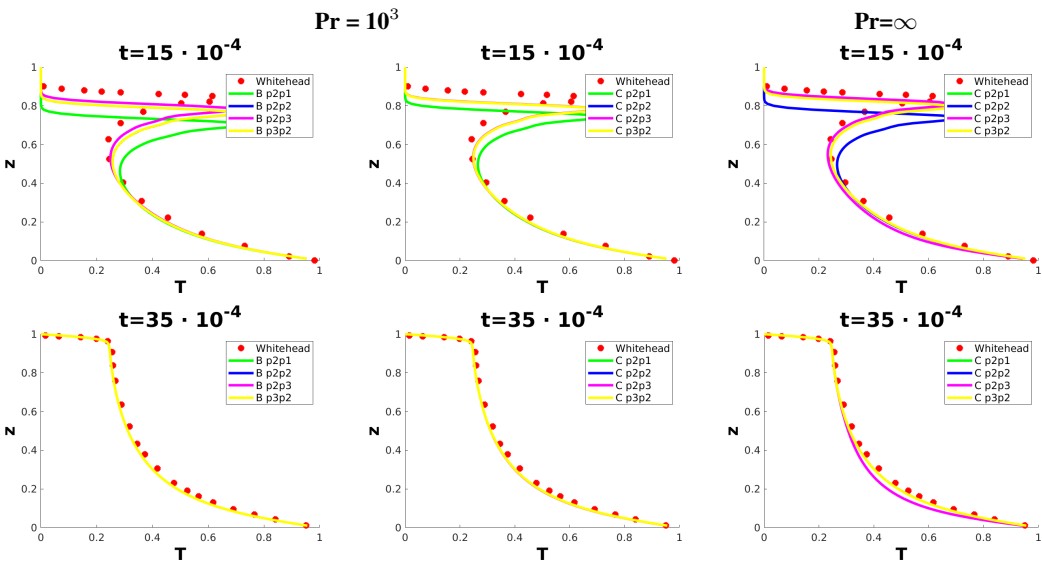

**Figure 13.** Vertical temperature profiles for two different times and different algorithms: p2p1 (green), p2p2 (blue), p2p3 (magenta) and p3p2 (yellow). At $t = 15 \cdot 10^{-4}$, p2p2 and p2p3 display the best accuracy for *buoyantBoussinesqPimpleFoam* (left), p2p2/p2p3/p3p2 for *convectiveFoam* (middle), p3p2/p2p3 for *convectiveFoamInf* (right). Other times display best accuracy with the p2p2 and p3p2 choices. All times are in unit of the diffusive time $\tau_\kappa$.





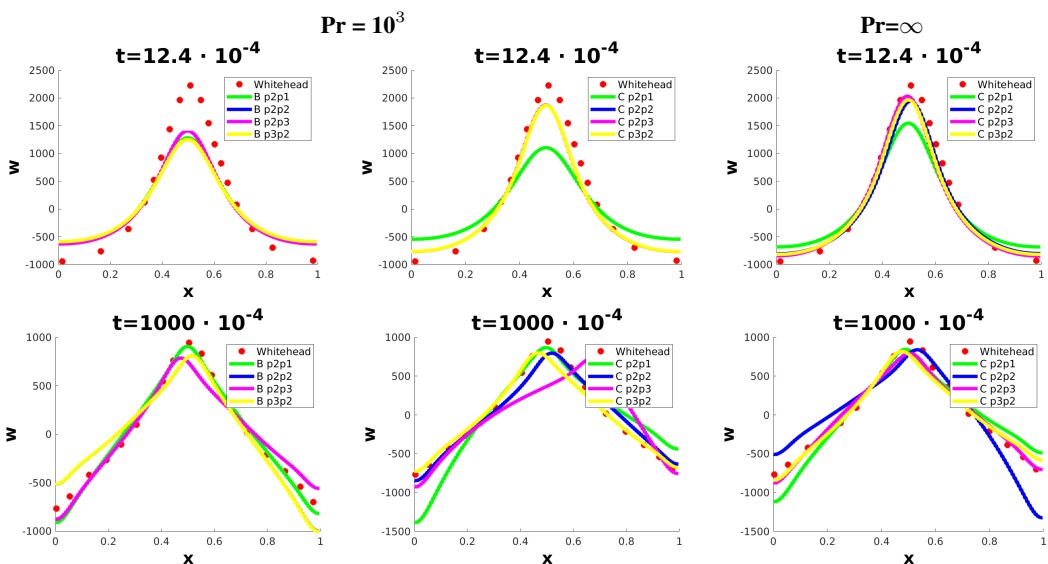

**Figure 14.** Horizontal profiles of vertical velocity for different algorithms. Profiles are taken as in Fig.8. Left two columns: *buoyantBoussinesqPimpleFoam* and *convectiveFoam* for $\text{Pr} = 10^3$; right column: *convectiveFoamInf* for $\text{Pr} \to \infty$. While *convectiveFoamInf* has better results at initial times, this is not true for $t = 1000 \cdot 10^{-4}$.

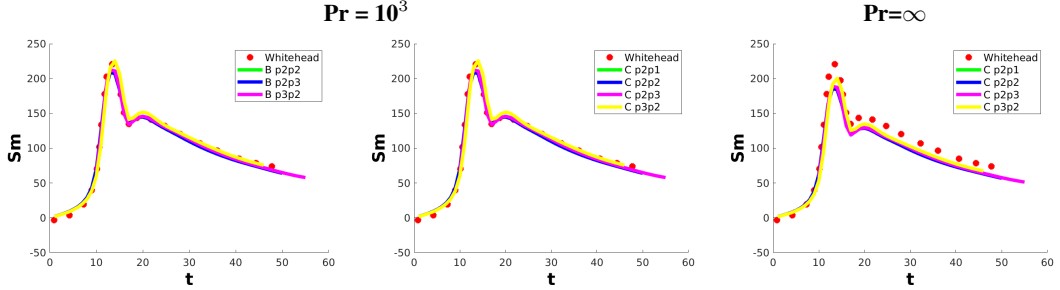

**Figure 15.** Streamfunction time evolution for p2p1 (green), p2p2 (blue), p2p3 (magenta) and p3p2 (yellow), in the interval $[0 - 50] \cdot 10^{-4}$. A better performance of the p2p2 algorithm is evident.





## 4.2 Analysis of the long term regime

With the same configuration of Sec. 4.1.2, we then analyze the statistically stationary regime until two thermal times. Referring to the long term results of Whitehead et al. (2013), we reproduced the temporal evolution of the streamfunction and compared

it with the original outcomes. As in the original work, we observed the emergence of two subsequent oscillating regimes with different frequencies, whose, unlike those of Whitehead et al. (2013), didn't lock in a steady oscillation of fixed frequency. To identify these oscillations we adopted the same approach of the original work, computing the heat flux (HF) and the streamfunction. The heat flux is defined as function of time:

$$\text{HF(t)} = N \times T(0.5, N-1, t) \tag{12}$$

where $T(x, z, t)$ is the temperature field, N the number of used level, the streamfunction is defined as:

$$\text{Sm(t)} = max(|\Psi|)(t) \tag{13}$$

where $\Psi$ is the proper streamfunction, but, for simplicity we will refer to Sm(t) as streamfunction. Results for both the Sm(t) and HF are reported in Fig.16, in which a shifting related to the increasing resolution (right panel) is observable as in the original work. A complete characterization of the oscillating phenomena has been performed evaluating the amplitudes of the oscillations in the long term regime. The main difference of our work is the absence of a long term regime with locked

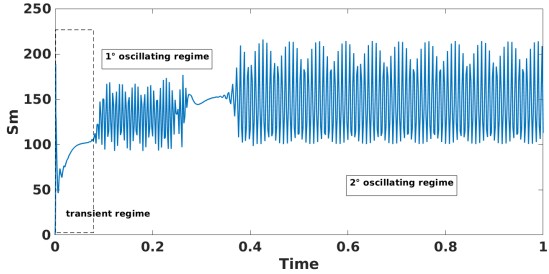

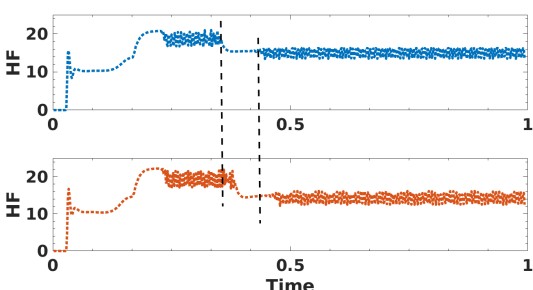

**Figure 16.** Left: several regimes of the process are identified: the transient, the first and the second statistically stationary regimes. Two different oscillations give rise to beats in the first and second oscillating regimes. Right: long term behaviour of the HF for increasing resolution and decreasing dissipation. In blue res= $64 \times 64$, in orange res= $128 \times 128$. As in the original work, the initial perturbation develops later for increasing resolution, shifting all the process ahead. The vertical lines indicate two times in which the shifting is particularly appreciable.


frequencies. To investigate this behaviour we ran simulations adopting increasing or decreasing dissipation (that is, increasing resolution). With the *Gauss Linear* scheme, the oscillations are completely damped, while, increasing resolutions or tolerance leave the oscillations active. For the moment we limit to report results for only two different configurations. The results, with relative details, are reported in Tab.4.





| Sim | Res | Tol | Ams1 | Af1 | T11 | T12 | Ams2 | Af2 | T2 |
|---|---|---|---|---|---|---|---|---|---|
| Whitehead et al. | $64 \times 64$ | - | 113.9 | 115.0 | 0.00152 | 0.00305 | 173.5 | 169.5 | 0.00269 |
| ConvectiveFoamInf | $64 \times 64$ | $3 \cdot 10^{-7}$ | 129 | 78 | 0.0060 | – | 149 | 116 | 0.0055 |
| ConvectiveFoamInf | $128 \times 128$ | $1 \cdot 10^{-10}$ | 131 | 80 | 0.0059 | – | 150 | 115 | 0.0077 |

**Table 4.** Comparison of several parameters for the $\mathrm{Pr} = \infty$, $\mathrm{Ra} = 10^6$ case. With the same convention adopted in Whitehead et al. (2013)(1 refers to first oscillation, 2 to the second): Ams1 = time average of Sm, Af1 = amplitude peak to peak. T11 = period of the first oscillation, T12 = the doubled period after the previous oscillation (not observed in our simulations), Ams2 = the steady value between the two oscillations (or the time average when a non steady behaviour is present), Af2 = amplitude of oscillations peak to peak, T2 = period of the second oscillation.

### 4.3   3D Benchmark

Three-dimensional simulations are run following the work of Sotin (1999). The simulation domain is a box with squared horizontal section $\left(\text{i.e. } \frac{L_z}{L_x} = \frac{L_z}{L_y} = \frac{1}{2}\right)$ with stress-free velocity BCs and fixed temperature on top and bottom and periodic BCs on the horizontal directions for both velocity and temperature. All simulations are run with the same configuration obtained from the previous analysis: CLB (*Convective, Linear, Backward*), p2p2 algorithm and $128 \times 128 \times 128$ grid points.

| Sim | Pr | Ra | Grid points | Sotin et al. | convectiveFoamInf |
|---|---|---|---|---|---|
| I | $\infty$ | $10^5$ | $128 \times 128 \times 128$ | $Q_t = 10.42(2)$ | $Q_t = 10.39(1)$ |
| | | | | $T_{mean} = 0.500$ | $T_{mean} = 0.552$ |
| II | $\infty$ | $2 \cdot 10^5$ | $128 \times 128 \times 64$ | $Q_t = 12.86(19)$ | $Q_t = 13.01(4)$ |
| | | | | $T_{mean} = 0.500$ | $T_{mean} = 0.500$ |
| III | $\infty$ | $5 \cdot 10^5$ | $128 \times 128 \times 128$ | $Q_t = 17.15(11)$ | $Q_t = 21.29(6)$ |
| | | | | $T_{mean} = 0.500$ | $T_{mean} = 0.471$ |
| IV | $\infty$ | $10^6$ | $128 \times 128 \times 64$ | $Q_t = 21.24(19)$ | $Q_t = 24.78(2)$ |
| | | | | $T_{mean} = 0.497$ | $T_{mean} = 0.507$ |
| V | $\infty$ | $10^6$ | $128 \times 128 \times 128$ | $Q_t = 21.24(19)$ | $Q_t = 29.74(12)$ |
| | | | | $T_{mean} = 0.500$ | $T_{mean} = 0.564$ |
| VI | $\infty$ | $3 \cdot 10^6$ | $128 \times 128 \times 128$ | $Q_t = 28.59(37)$ | $Q_t = 44.71(24)$ |
| | | | | $T_{mean} = 0.500$ | $T_{mean} = 0.561$ |

**Table 5.** Columns 1-4: configuration for each 3D simulation. Columns 5-6: comparison of heat flux and mean temperature evaluated at the stationary state for the original data and *convectiveFoamInf*. For each value of the heat flux is reported the mean value and the relative fluctuation, for the temperature field, the value averaged over the volume.

We reproduced some cases selected from the original work and validated our solver comparing the statistically stationary behaviour of the flux at the top boundary $Q_t$ and the mean temperature in the volume of domain. As shown in columns 5-6 of Tab.5, the *convectiveFoam* results are close to those of Sotin (1999) and catch the increase of the heat flux and fluctua-



tions for higher Rayleigh numbers. The values are closer to each other for lower values of the Rayleigh number, while some discrepancies emerge as the turbulence level increase. Future simulations will further address this discrepancy.

### 4.3.1 Performances

Parallel efficiency has been estimated by evaluating the strong scaling, that is how the computation time varies with increasing number of cores, keeping the problem size fixed. We defined the speedup as the ratio:

$$Sp = \frac{T(1)}{T(n)} \tag{14}$$

where $T(1)$ and $T(n)$ are the execution times on one and $n$ cores, respectively. Results of the simulations are reported in Fig.17 where the performance significantly decreases for simulations with more than 16 cores, corresponding to about 16000 cells/core. This result is consistent with Cerminara et al. (2016) and Paronuzzi Ticco S.V. (2016) which used an analogous numerical structure (i.e. in the FV method) reaching the best efficiency around 13000 cells/cores. In this work, due to the simplification of the model (no advection of the velocity field), this performance is slightly increased to about 16000 cells/cores. All simulations have been run on the Marconi cluster at CINECA ($1 \times 68$-cores Intel Xeon Phi7250, KNL, at 1.4 GHz).

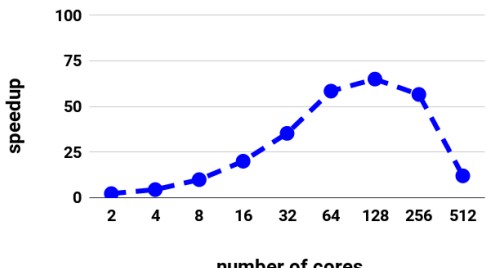

**Figure 17.** Speed up law for the strong scaling problem. The problem size is fixed to $128 \times 128 \times 128$ cells. At 128 cores (maximum speed up) the cells per processor ratio is 16000.

### 4.4 Further tests and portability

For completeness, additional test considering a relaxation parameter have been done. Results with relaxation exhibit a smoothing effect for both the temperature field and streamfunction with respect to those without relaxation, thus introducing a non acceptable deformation of both finite and infinite-Pr number results.

*ConvectiveFoamInf*, originally developed on OpenFOAM5, is available also on OpenFOAM7.

The relaxation and the portability tests, have been performed monitoring the same observables of the previous benchmark: the vertical temperature profile and isolines of temperature and streamfunction. For all the described studies results are available under request, as for the complete results for the benchmark.



## 5 Conclusions

We built and tested *convectiveFoam*, a new solver dedicated to the simulation of infinite-Pr number fluids, using the Open-
FOAM toolkit. The development of the new solver moved from very high, finite-Pr number to the infinite-Pr case. We called
this different variants *convectiveFoam* and *convectiveFoamInf* respectively. We tested the solver reliability by quantitative com-
parisons with the 2D simulations described by Whitehead et al. (2013) and the 3D simulations reported by Sotin (1999). A
satisfactory agreement between our results and such cases was found. In particular, the analysis of the 2D cases indicates that:

i) Moving the temperature equation into the PISO loop improved the performance, by reducing significantly the number of
required PIMPLE loops to achieve convergence. The $T_{eq}IN$ solver is about 3-5 times faster than $T_{eq}OUT$.

ii) Resolution higher than $128 \times 128$ did not considerably improve the results, with the exception of highly dynamic times for
which a finer grid can give a more accurate description in all the domain. As in Whitehead et al. (2013), the bigger discrepancies
are present for the $64 \times 64$ resolution.

iii) Concerning schemes, even if the LLB and LLE schemes produced good results, the LB choice, combining the spatial
*Linear* and temporal *Backward* schemes, gave the best compromise between time simulation and matching with original results.

iv) Although we discarded some algorithms because of their inconsistency with original results, several combinations of
PISO and PIMPLE loops gave good results in terms of reproducibility. The p2p2 algorithm was chosen as the optimal com-
promise between simulation cost and agreement with the previous results.

v) Further studies performed with and without relaxation parameter revealed that the use of relaxation gave results that were
not consistent with the reference study. Further investigation is needed to further explore the origin of this behaviour.

For the 3D case, comparing our results with Sotin (1999) we found that:

vi) The *convectiveFoamInf* solver was able to reproduce main results of Pr $\rightarrow \infty$, isoviscous fluid simulations even if a more
accurate study on resolution is necessary to avoid non-physical results.

vii) Strong scaling analysis showed a maximum speedup around 16000 cells/cores.

viii) Portability was evaluated for the latest release OpenFOAM7.

We remark that in this work we analyzed idealized models. This step is just the first of a longer project: further developments
should consider rheological aspect of the Earth mantle such as the temperature/pressure viscosity dependence, which still has
unsolved numerical issues in 3D domains (Khaleque et al., 2015). Also, analysis of both Newtonian and non Newtonian
viscosity has to be considered.

*Code availability.* The code developed in this work is available for collaboration or download at *https://doi.org/10.5281/zenodo.3718556*.
At the same link, also some examples and a general description of the code are available.

*Author contributions.* S.L. performed the numerical simulations and analyzed the data, conducted literature research, and wrote the first draft
of the paper. M.C. and S.L. developed the codes modifying the OpenFOAM libraries and prepared the simulations. M.C., M.d.V., T.E.O. and





A.P. supported the definition and the analysis of the results. A.P. supported the definition of the numerical experiments and their implications.
A.P., T.E.O. and M.d.V. conceived the project. All authors contributed in the design of the study, discussed the results, and commented on
the paper.

*Competing interests.*   The authors declare no competing interests.

*Acknowledgements.*   Most of simulations have been run on *Laki*, the high-performance computing resources of INGV (Pisa) to which the
first author is very greatfull. We also acknowledge the CINECA for providing high-performance computing resources.
This work took advantage of the CINECA infrastructure through the ISCRA projects: IsC SIPRACo2 and IsC SIPRACo3.
This work was partially supported by the ASI-INAF Agreement No. 2018-25-HH.0 "Scientific activities for JUICE phase C/D."



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
