# Peer review of "ConvectiveFoam1.0: development and benchmarking of a infinite-Pr number solver."

_Geoscientific Model Development, 2020_

## Referee Comment (RC1) · Anonymous Referee #1 · 1 May 2020

There are a number of reasons why I did not feel that there was anything here that was worth publishing in peer-review literature.

1. From the point of view of computational fluid dynamics, the infinite Prandtl number limit should pose fewer problems that the finite Prandtl number case, since the highly non-linear inertial terms in equation (2a) become negligible. The overall problem is still non-linear of course, but there is nothing intrinsically difficult about the infinite-Pr limit, and it has already been considered by many others.

2. From the point of view of geoscience, the constant viscosity case is of limited interest, and the numerical challenge is rather how to deal with a temperature-dependent viscosity that varies by many orders over the temperature ranges that are of practical interest.

[Figure]

Whilst I can understand that the authors are developing a numerical model on a new software platform (OpenFOAM) from scratch, and that this process needs benchmarking, it is debatable whether this needs to be documented in peer-review literature (perhaps at best a technical report, or within a Ph.D. thesis).

———————————————————

---

## Referee Comment (RC2) · Anonymous Referee #1 · 2 May 2020

The infinite-Pr limit was considered at length in Vynnycky, M. & Masuda, Y., Rayleigh-Bénard convection at high Rayleigh number and infinite Prandtl number: asymptotics and numerics, Phys. Fluids 25 (2013) Article number 113602

Refs. 2-9 therein consider numerical solutions to the problem, whereas Vynnycky & Masuda themselves attempt to reconcile their own numerical and asymptotic solutions.

Other relevant references (see Clarivate Analytics Web of Science) may be:

Numerical Simulation of Two-Dimensional Rayleigh-Benard Convection By: Grigoriev, Vasiliy V.; Zakharov, Petr E. Conference: 8th International Conference on Mathematical Modeling (ICMM) Location: NE Fed Univ, Yakutsk, RUSSIA Date: JUL 04-08, 2017 Sponsor(s): Ammosov NE Fed Univ PROCEEDINGS OF THE 8TH INTERNA-

TIONAL CONFERENCE ON MATHEMATICAL MODELING (ICMM-2017) Book Series: AIP Conference Proceedings Volume: 1907 Article Number: UNSP 030031 Published: 2017

Numerical simulation of two-dimensional Rayleigh-Benard convection in an enclosure By: Ouertatani, Nasreddine; Ben Cheikh, Nader; Ben Beya, Brahim; et al. COMPTES RENDUS MECANIQUE Volume: 336 Issue: 5 Pages: 464-470 Published: MAY 2008

---

## Referee Comment (RC3) · Anonymous Referee #2 · 6 May 2020

This manuscript decribes a numerical solver for convection at very large Prandtl numbers based on OpenFOAM platform. The development of a parallel code for convection problems (both in 2D and 3D) on open source platform is important for the numerical research in geophysical convection and I think that the present manuscript deserve publication. The main limitation of the manuscript is that, in the present version, it is more a technical report than a scientific paper. Most of the manuscript is devoted to technical details which are not really relevant for future use of the code. My suggestion is to move most of these discussions into appendices and leave in the article the main features of the code only. Moreover, for a publication in GMD I would like to see more discussions of the results presented in Sections 4.2 and 4.3 with physical interpretations. Section 2 should also be improved. The last term in (2a) disappears in (3a). In

any case, why rotation is introduced if in (4) it is not present?

---

## Author Comment (AC1) · 3 Jul 2020

1. The innovative part of this work is strictly linked to the mathematical and the computational aspects of the problem.

The mathematical structure of the momentum equation changes its nature in the $Pr \rightarrow \infty$ limit (treated in Sec 3.1), and, indeed, different approaches are required to solve the two different kinds of second order partial differential equations. In the infinite Pr case, the momentum equation becomes a contraint (a diagnostic relationship).

Given this, the strength of this work consists in maintaining the original diagonal dominant structure of the OF solver, suited for the finite-Pr cases, to reproduce results in agreement with the $Pr = \infty$ case. This agreement is reached, once the convergence

criteria are satisfied.

2. As mentioned, the infinite-Pr solver is still lacking in the OpenFOAM community and the code benchmarking can't disregard from steps in which a constant viscosity is assumed.

The opinion of the authors is that this fundamental step must be made available to the community, by virtue of the open source nature of the platform. Our opinion is that this work should be documented on GMD because we think it meets the scientific purpose of the journal: GMD "...is a not-for-profit international scientific journal dedicated to the publication and public discussion of the description, development, and evaluation of numerical models of the Earth system and its components...".

Finally, as noted by the referee, since from the geoscience point of view a temperature-dependent viscosity is the more interesting application at now, *"...the model is conceived to include future applications with non-Newtonian viscosities (dependent on temperature and pressure), multi-phase and multi-component flows..."* (as reported in the Introduction).

―――――――――――――――――

---

## Author Comment (AC2) · 3 Jul 2020

1. As suggested by the referee, the structure has been modified to make it more readable. Three appendices now include the more technical parts of the work, keeping the mathematical aspects and the main code features unchanged. In particular, numerical details and related considerations are reported in appendix A, the main part of the benchmark can be consulted in appendix B while further studies can be found in appendix C.

2. Section 4.2 and 4.3 have been extended with more physical interpretations and related references. In particular we referred to the following:

- *Krishnamurti, R.: On the transition to turbulent convection. Part 1. The transition from two-to*

*three-dimensional flow, JFM, 1970a.*

- *Krishnamurti, R.: On the transition to turbulent convection. Part 2. The transition to time-dependent flow, JFM, 1970b.*

- *Krishnamurti, R.: Some further studies on the transition to turbulent convection, JFM, 1973.*

- *Busse, F.: The oscillatory instability of convection rolls in a low Prandtl number fluid, JFM, 1972.*

- *Busse, F. and Whitehead, J.: Oscillatory and collective instabilities in large Prandtl number convection, JFM, 1974.*

- *Busse, F. H. and Whitehead, J.: Instabilities of convection rolls in a high Prandtl number fluid, JFM, 1971.*

Section 2 has intentionally left short for readability reasons.

3. Considering rotation in Eq.2a, our intention was to develop the more general treatment to show that, adopting $Ek \approx 10^{12}$, rotation effects are negligible in Earth Mantle convection. In this new version, we removed the rotation terms to simplify the presentation.

---

## Author Comment (AC3) · 3 Jul 2020

Some results for the 3D benchmark has been updated with outcomes of newer simulations (see Table 3).
* * *